# Microplastics in Ship Sewage and Solutions to Limit Their Spread: A Case Study

**Renate Kalnina** [1,*] **, Ieva Demjanenko** [1] **, Kristaps Smilgainis** [1] **, Kristaps Lukins** [1] **, Arnis Bankovics** [1] **and Reinis Drunka** [2]

[1] Riga Technical University Latvian Maritime Academy, Flotes Str. 12 k-1, LV-1016 Riga, Latvia
[2] Faculty of Material Science and Applied Chemistry, Institute of Materials and Surface Engineering, Riga Technical University, Paula Valdena, 3, LV-1048 Riga, Latvia
[*] Correspondence: renate_kalnina@inbox.lv

**Abstract:** The case study presented in the paper is the first in the field to find microplastic (MP) particles in both grey water (GW) and post-treatment sewage (TS) samples, which can also be legally discharged into specially protected areas. Compiling a data set of 50 water samples collected from the GW and TS samples of 5 transport ships involved in the case study, we show that the mean number of separated microparticles in the GW samples $n = 72$ particles per litre, and in the TS samples $n = 51$ particles per litre. Of the 614 separated particles, the most common were fibres $n = 285$ (46.4%), followed by other (various) hard particles $n = 226$ (36.8%) and soft particles $n = 104$ (16.8%). Attenuated total reflectance Fourier transform infrared spectroscopy (ATR FT-IR) identification was mainly in the form polyethylene (PE), polypropylene (PP) (solid particles and films), polyesters, polyamides, and acrylic fibres. Scanning electron microscopy (SEM) and energy-dispersive X-ray spectroscopy (EDS) analysis revealed ecotoxic chemical elements on the surface of these particles. Our results show that the sewage treatment facilities installed on ships need to be improved, and we developed a solution for this. The findings of the case study certainly deserve further attention and serve as an impetus for research on the improvement of ship wastewater treatment facilities.

**Keywords:** microplastics; grey water; treatment sewage; sewage treatment plant; removal efficiency

## 1. Introduction

About 75% of the earth's surface is covered by water biomass. It is an essential natural resource that provides countless life cycles while providing various environmental, economic, social, and cultural services [1]. The use of this critical natural resource in international freight transport has increased significantly in recent decades. It is estimated that 11.08 billion tons, or 90% of world international cargo trade, is carried by ships [2]. Despite the significant contribution of maritime transport to the freight sector, there are concerns about the environmental impact of operational and accidental spills from ballast, run-off, waste, and sewage. Ship effluents are mistakenly considered to be the same as on land. Unfortunately, they are much more concentrated than inland wastewater, and their concentration is further increased by vacuum collection systems [3]. Domestic wastewater is treated in wastewater treatment plants, which remove about 90% of its microplastics. However, they have been identified as a significant source of microplastics for the marine environment [4]. At present, there is a lack of understanding and research into the possibility of microplastic contamination in ships' sewage.

Artificial polymer materials, which we call plastics, have become indispensable in our daily lives. Versatility has been determined by the excellent properties, low cost, and excellent design of plastics, replacing many traditional materials such as paper, glass, and metal [5]. However, plastic products do not have the same application or service life. It may be less than 1 year or more than 50 years. Plastics can form both the complete as

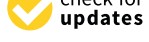



well as part of the final consumer product, and at the end of the life cycle, they become waste that should be collected and recycled. Even though the amount of waste sent for recycling has doubled since 2006, 25% of it is sent to landfills [6] or naturally enters the marine environment from land [7]. It is estimated that it accounts for about 80% of the world's anthropogenic plastic debris in the seas and oceans [4].

Plastic debris in the sea can be divided into macro-, meso-, micro-, and nanoplastics [5]. Marine plastic debris in the range from 1 nm to 5 mm is classified as microplastics (MPs) [8,9]. They have two sources of origin, namely primary and secondary. Primary MPs are those that were initially produced as small particles. Such particles are present in many consumer products as exfoliating/abrasive products for personal care and cosmetics [10–12] or as an ingredient in synthetic textiles [13]. Larger plastic objects that are intentionally or unintentionally discharged or lost at sea and, as the result of physical or photo-oxidation collapse, produce particles called secondary MPs [5,8]. Both MPs have been found in the marine and ocean environment, but primary MPs exceed secondary MPs' emissions from mismanaged waste. Studies show that any MPs can travel long distances [14] and that the seafloors are the last site of these particles to degrade slowly and become underwater landfills [8]. Many plastics can persist in the marine environment for hundreds or even thousands of years [15] and can absorb toxic substances and pathogens on their surface, causing potential damage to the ecosystem and, through accumulation in the food chain, can affect human health [16–18].

A global study of seven world regions revealed the absolute value of MP emissions per region. It ranges from 134 to 281 K ton/year but varies significantly in each region from 110 to 750 g/year per capita [19]. Concentrations of MP emissions correlate with population density [20], per capita income improvements, and various human activities such as plastics production, shipping, fishing, port operations, and tourism or disordered MP emissions distribution systems such as wastewater treatment systems [9,19,21,22]. Therefore, the above proportional distribution of 20% for marine sources and 80% for terrestrial sources is transparent [23], as it is estimated that about 37% of MPs enter the marine environment through sewage. In addition, the understanding of MPs' contributions to shipping, fisheries, and aquaculture is limited due to limited information sources and scientific research.

According to a review by the International Maritime Organization (IMO), MPs can be caused by tools and equipment intentionally abandoned or accidentally lost from fishing vessels and aquaculture facilities, as well as paint and marine coating particles, offshore effluents, and transported ballast water [24]. Ship effluents are one of several sources of waste discharged into the marine environment from merchant ships and passenger and cruise ships [8]. Due to natural bacteria, the open sea or ocean water generally can assimilate and deal with untreated ship sewage. However, ship effluents contain not only nitrogen and phosphorus but also bacteria, viruses, detergents, and heavy metals, which can have irreversible effects on the marine environment. Therefore, the provisions of Technical Annex IV to the International Convention for the Prevention of Pollution from Ships (MARPOL 73/78) govern the transport and discharge of wastewater over a specified distance from the nearest land unless otherwise specified. In turn, states must ensure adequate provision in ports and terminals for wastewater reception from ships without causing delays [25–27]. Ships' sewage can be divided into two categories: "black water" (BW) and "grey water" (GW). BW refers to wastewater and other wastes from toilets of any kind (including medical premises and spaces containing living animals), urinals, or other wastes mixed with the above. GW refers to drains from washbasins, galley sinks, showers, laundries, bathtubs, and washbasin drains but does not include BW (IMO). Studies show that GW can range from 105 L per person per day to 222 L per person per day. The amount depends on vessel type, persons on board, and other variables [28]. An average of 61% of GW comes from the cabin area, 25% from the kitchens, and 14% from the laundry [29]. Technical Annex IV to MARPOL 73/78 does not regulate the composition and discharge of GW. According to Technical Annex IV, onboard wastewater may be managed as follows:

1.  BW and GW/galley waters can be collected in a storage tank and discharged to the port for further treatment.
2.  BW can be ground, disinfected, and discharged directly into the sea with GW/galley water.
3.  GW/galley water can be kept in dedicated tanks and periodically discharged directly into the sea.
4.  GW can be cleaned and used for a toilet flushing system [30].

As a result, researchers are concerned about the lack of monitoring and information on the content of pollutants such as MP, and GW, which could be a significant source of marine pollution from ships. This study aimed to (1) investigate the presence of microplastics in ship effluents, namely GW and treated sewage (TS) samples collected from five different transport vessels; (2) classify the MPs by size, colour, and shape; and (3) and identify and (4) discuss possible sources of MP in the ship's sewage, and (5) suggest a possible solution for the collection of MPs in a ship's sewage treatment plant. The study results support the hypothesis about GW and TS as a source of MP pollution. The results of this study may be of interest to researchers working in a similar thematic area and the maritime industry to increase the eco-efficiency of ships and policymakers in the marine environment field.

## 2. Materials and Methods

### 2.1. Study Area and Sample Collection

The study was conducted from June 2020 to December 2021, involving ships sailing in the Baltic Sea. GW and TS samples of ships were collected for physicochemical analysis from five transport ships when they called at the ports of Klaipeda, Liepaja, Riga, and Helsinki. The crews of the transport ships consisted of 17 to 23 persons. It is estimated that in an average of 24 h, 1 person produces around 105L GW and 25L BW.

The transport ships included in the study were equipped with sewage treatment plants. It is essential to point out that the sewage treatment plant's operating principle was based only on BW treatment. Consequently, the GW/galley water and the treated BW are discharged directly into the sea. However, if the ship is sailing closer than 12 nautical miles from the nearest land or is in port or the Baltic Sea area, the GW/galley water is fed into the sewage treatment plant and treated with the BW. After treatment, wastewater can be discharged into the Baltic Sea or the port's water area [31]. Therefore, two types of samples, namely 1 L of GW and 1 L of treated sewage containing both GW and BW, were collected from each transport ship to obtain comparable data on microplastic content.

In the absence of a standardized method for sampling microplastics from ship sewage, the United States Environmental Protection Agency (US EPA) sampling advice was followed to obtain a representative sample of the entire ship's GW discharge. According to the principle of proportionality, GW samples were collected in 200 mL glass bottles from several water outlets (Figure 1). Then, they were combined in a special 1 L glass bottle. The bottle was capped tightly and labelled with the sampling location, date, and time. All samples were delivered to the laboratory within 24 h and stored at 4 °C before analysis.

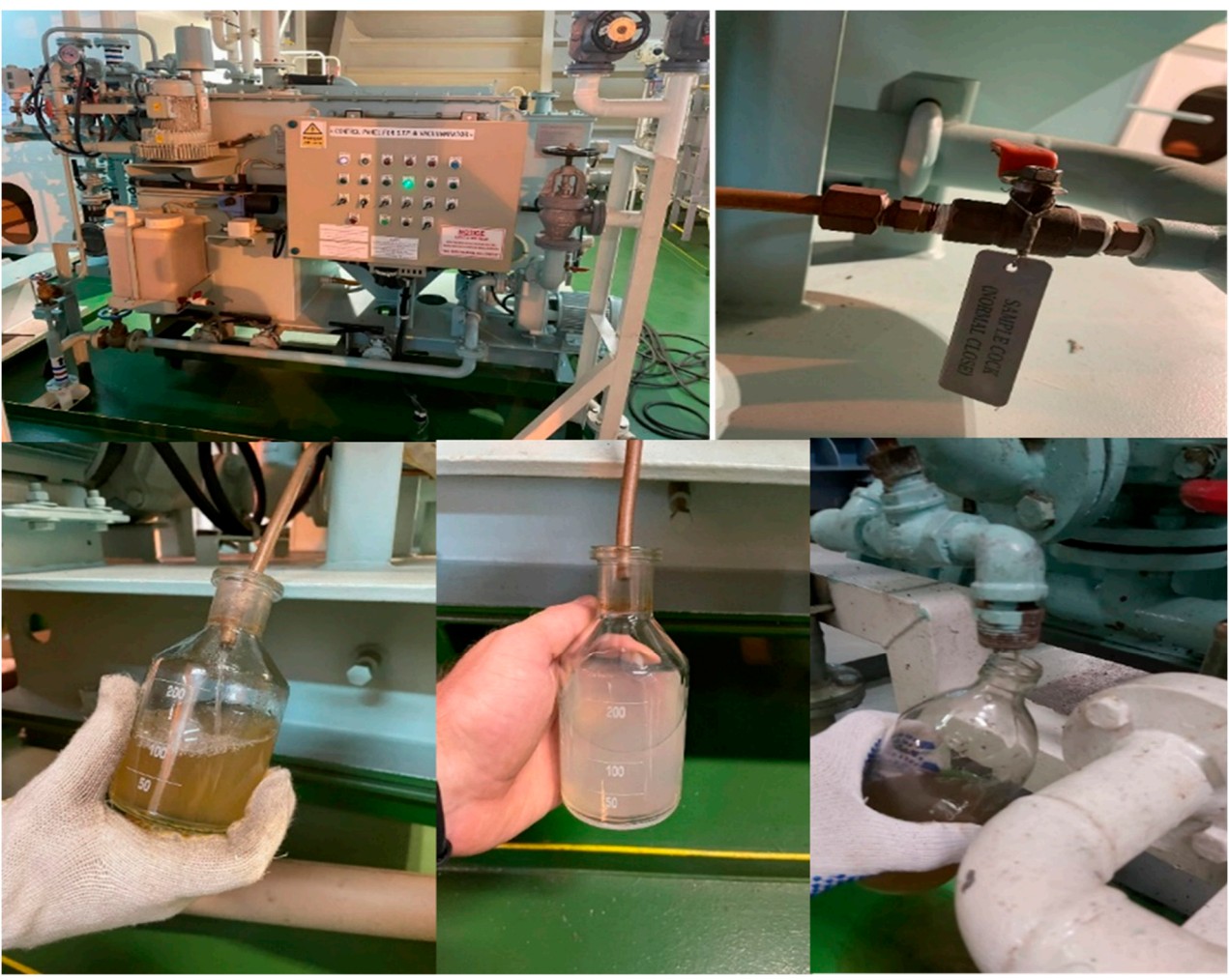

**Figure 1.** Ship "grey water" sampling from the transport vessel.

*2.2. Grey Water Sample Processing and MPs Identification Methods*

Our analytical procedure for the preparation of GW samples for the identification of MPs (shown in Figure 2) was inspired by the U.S. National Oceanic and Atmospheric Administration (NOAA) Laboratory Methods for Microplastics in Various Matrices in the Marine Environment [32]. However, in this case, a modified NOAA method was used to prepare, isolate, and identify MPs from GW samples and was developed based on the recommendations [13,33–35].

GW sample preparation and MP purification and analysis consisted of three steps. In the preprocessing, GW samples were concentrated on a rotary evaporator to a volume of 200 mL and transferred to a 400 mL glass beaker, and the rotary flask was rinsed with 50 mL of ultrapure water and added to the sample; then the organic matter was digested with 30 mL 30% hydrogen peroxide (Sigma Aldrich, Taufkirchen, Germany). Beakers were then placed on magnetic stirrers, heated to 50 °C, and left to react for 15 min. Additional 10 mL of $H_2O_2$ were added after 30 and 90 min, and finally, the mixture was left to react for 20 h. The digestion time was chosen as optimal for ensuring the complete removal of organic matter in the most challenging samples without affecting the integrity of microplastics. Then, 15 g of sodium chloride (NaCl) (99.5%, Sigma Aldrich, Taufkirchen, Germany) was added to allow polymers with a wide density range to float and separate, and then the sample was allowed to cool to room temperature.

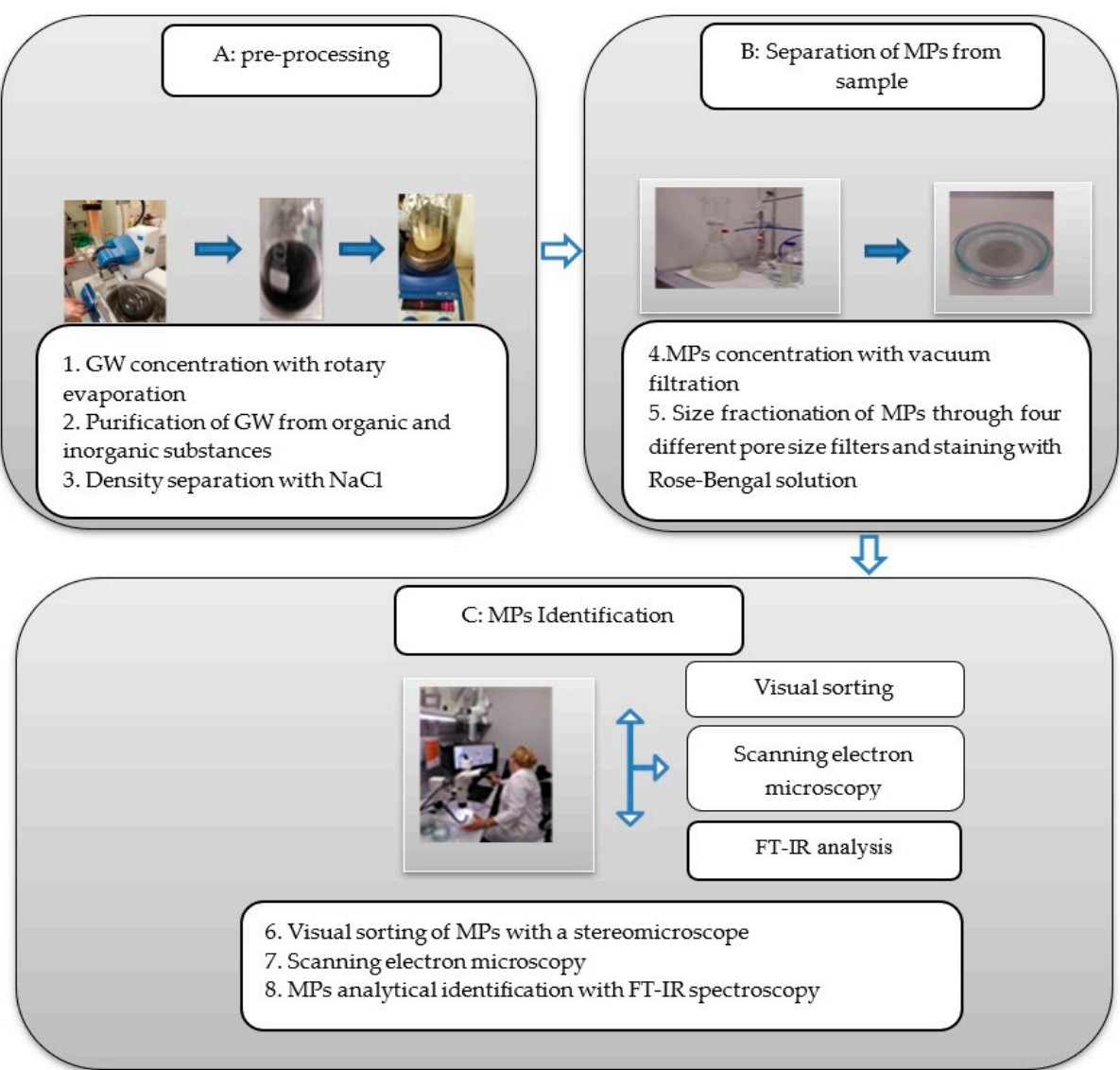

**Figure 2.** The visual representation of analysis procedures.

In the second step, the cooled sample was filtered through a stainless-steel sieve with a mesh size of 300 and 100 μm to remove the largest solid particles. The obtained filtrate was vacuum filtered through glass fibre material filters (Sigma-Aldrich, Darmstadt, Germany) with pore sizes of 51 and 0.7 μm, respectively. All filters were stored in precleaned glass petri dishes before the following procedure.

On the other hand, to separate the nonplastic particles still possibly remaining in the samples, e.g., natural cotton fibres, which are optically very similar to synthetic fibres, a staining method was applied using rose bengal solution (4,5,6,7-tetrachloro-2',4',5',7'-tetraiodofluorescein Sigma-Aldrich, 95 % dye content). The wet filter (Sigma-Aldrich, Taufkirchen, Germany) was subjected to rose bengal staining according to [36]. Each sample was covered with 2 mL of 200 mg/L rose bengal solution on the surface of the mesh and left to react for 5 min at room temperature. Finally, the dye was washed with ultrapure water, and the mesh was dried in the drying oven Nabertherm L9/B (Nabertherm manufacture, Lilienthal/Bremen, Germany) equipped with the proportional-integral-derivative P330 controller at 60 °C for 15 min. After the sample was dried, the stained particles were isolated under a stereomicroscope (Nikon SMZ 800N with a zoom range of 1–8x, 80X and a high resolution of 640 LP·mm$^{-1}$) (Nikon Corporation, Tokyo, Japan), and a hot needle test

was performed according to the procedure adopted from [32] to verify that the separated particles were not microplastics.

In the third step, the laboratory assistant separated MPs from each sample with the help of a stereomicroscope (Nikon SMZ 800N). The MPs were counted and classified according to their size, colour, and type, including irregular fragments, films, and granular fibres. To further characterize the MPs, representative particles from each group of particles was collected by the operator, immerse in distilled water, and stored until analytical tests were performed. Microplastics of different sizes and shapes and polymer types are challenging to identify using a single analytical method [37]. For the further characterization and elemental analysis of the surface of MP particles, scanning electron microscopy (SEM) with energy-dispersive X-ray spectroscopy was used, which provided precise and high-magnification images of the surface texture of the particles and the elemental composition of the same object.

Before analysis, the MPs containing liquid samples were filtered through a cellulose acetate filter (Sigma-Aldrich, Darmstadt, Germany) with a mesh size of 0.22 μm. The filter was dried in the drying oven Nabertherm L9/B equipped with the proportional-integral-derivative P330 controller at 105 °C for four hours. Subsequently, MP samples were cooled down to room temperature inside the desiccator. The operator covered each piece of MPs with a 15 nm thin layer of gold to provide higher surface conductivity of organic and plastic materials. Obtained MP samples were scanned with the help of scanning electron microscopy (SEM, Tescan Mira/LMU, Brno, Czech Republic) at 15 kV acceleration voltage. The composition of MPs was analysed with energy-dispersive X-ray spectroscopy (EDS). For identification purposes, MP particles were analysed by attenuated total reflectance (ATF) Fourier transform infrared (FT-IR) spectroscopy (Bruker HTS-XT Vertex 70). Spectra were obtained at a resolution of 4 cm$^{-1}$ by averaging 64 scans in the wavenumber range from 400 cm$^{-1}$ to 4000 cm$^{-1}$. FT-IR spectroscopy provides information on the specific chemical bonds of particles. The obtained spectra were compared with a comprehensive commercial spectral library (Bruker ATR-FT-IR Complete Library, which includes more than 26,000 reference spectra obtained in 2019) to identify MP particles with the highest confidence. A positive match between the sample and library spectra was scored when at least 70% similarity was obtained. The sample preparation and identification procedure for the treated sewage (TS) consisted of the three steps described above as for the GW samples.

### 2.3. Statistics

Confidence intervals (CI) were calculated at the 95% level with at least three replicates for each typology. FT-IR analysis was used to identify microplastics, while Pearson correlation was used to assess the compliance of the samples with the databases or standards.

### 2.4. Quality Assurance and Quality Control

The following measures were implemented throughout the sample processing and analysis to prevent contamination with microplastics in the laboratory from external sources: all laboratory personnel were required to wear cotton lab coats with buttons down the front. They always had to wear powder-free, nitrile-coated gloves when performing any work in the lab. All glassware was thoroughly rinsed three times with deionized water, covered with aluminium foil, or kept in a metal box. Sample processing was performed under a laboratory fume hood. Negative control samples were also processed at all stages of sample processing to detect possible plastic contamination from the laboratory. No microplastics were observed in any negative control samples, indicating that the experimental procedure and sample treatment reagents did not introduce plastic contamination.

## 3. Results

### 3.1. Microplastic Visual Sorting Analysis Results

According to typology, microparticles were first divided into three morphotypes. The first morphotype included solid particles such as irregularly formed fragments, beads, or

spheres, labelling these particles as another hard form. The second morphotype included soft films. The third morphotype consisted of fibres. In the study, the authors recognized the fibres as microparticles with a cylindrical shape and length to diameter ratio > 3 as defined by the European Chemicals Agency [38].

Typologically different microplastic particles were identified in all ship GW and TS samples. Fibres were found most often. Their number was reduced by separating *n* = 110 natural fibres dyed using rose bengal dyeing and *n* = 51 fibres that were composite polymer. Rose bengal dyed the natural polymer fibres for these fibres, but the synthetic polymer fibres were not dyed (Figure 3). The results of fibre dyeing show that they can be a composite or semi-synthetic material that is produced from both natural and synthetic fibres. The authors of this paper plan to determine the composition of the separated fibres in a future case study.

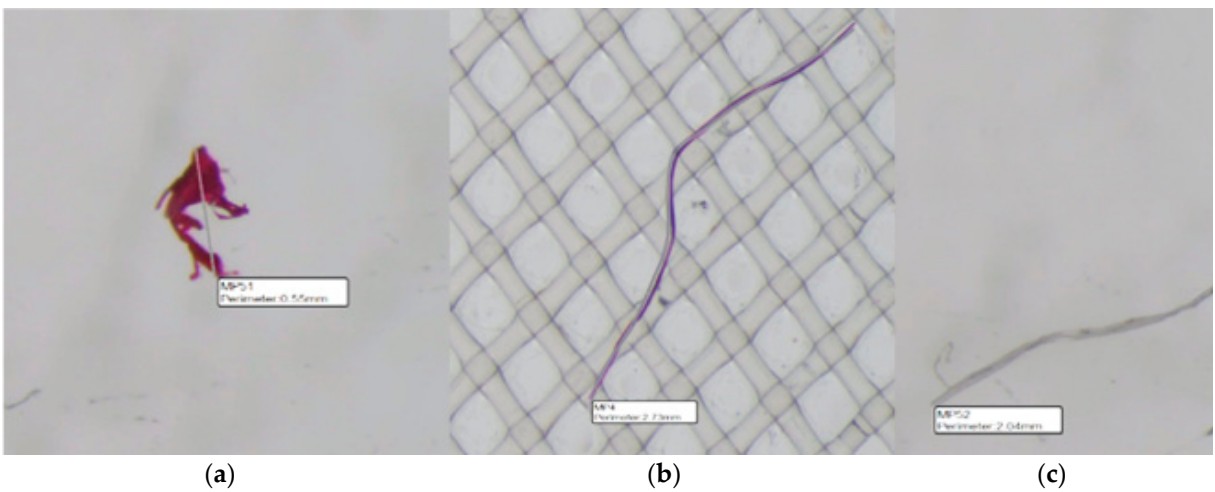

(**a**)  (**b**)  (**c**)

**Figure 3.** Particles after staining with rose bengal (**a**), partially dyed semi-synthetic fibre (**b**), and synthetic fibre (**c**).

A total of 614 particles were separated from 5 GW samples with a total volume of 5 L and 5 TS samples with a total volume of 5 L. The average number of separated microparticles in the GW sample was *n* = 360, but the TS sample contained *n* = 254 particles. The average number of separated microparticles in GW samples was *n* = 72 particles per litre, and in TS samples, *n* = 51 particles per litre.

Among the 614 separated particles, the most common were fibres *n* = 285 (46.4%), followed by other hard particles *n* = 226 (36.8%) and soft particles *n* = 104 (16.8%). A total of 10 different colours formed the visually separated microparticles. Fibres were predominantly light/grey in *n* = 98 (34.4%). Different solid-shaped particles were separated in different colours. Two colours dominated, namely red *n* = 72 (31.9%) particles and white *n*= 67 (29.8%) particles. Soft forms or films *n* = 15 (14.8%) were transparent, as shown in Figure 4.

The retained microparticles were divided into 2 categories according to the mesh size range of the filter mesh, namely 0.7–100 μm and 100–300 μm. When filtering both GW and TS samples, more shaped fibre particles were retained on the filters with the smallest mesh size. In the range 0.7–100 μm, *n*= 211 (74%) fibres were separated. On the other hand, hard particles of various shapes were separated by solid-shaped particles *n*= 88 (39%) and soft-shaped particles by *n*= 41 (40.6%). Fibres that exceeded the filter mesh size were identified on many filters. For example, a 2.04 mm long fibre was identified on the surface of the 0.7 μm filter mesh screen. This can be explained by fibre morphology, i.e., the large length-to-diameter ratio difference allows them to pass lengthwise through filter screens even with a tiny mesh size. The identified MP particles ranged in size from 10 μm to 3.68 mm. The average size of the identified particles was 1.26 mm. The fibres

had the largest length range from 30 μm to 3.58 mm. The distribution of the number of microparticles removed from the samples by morphotypes on the filter mesh screens can be seen in Table 1.

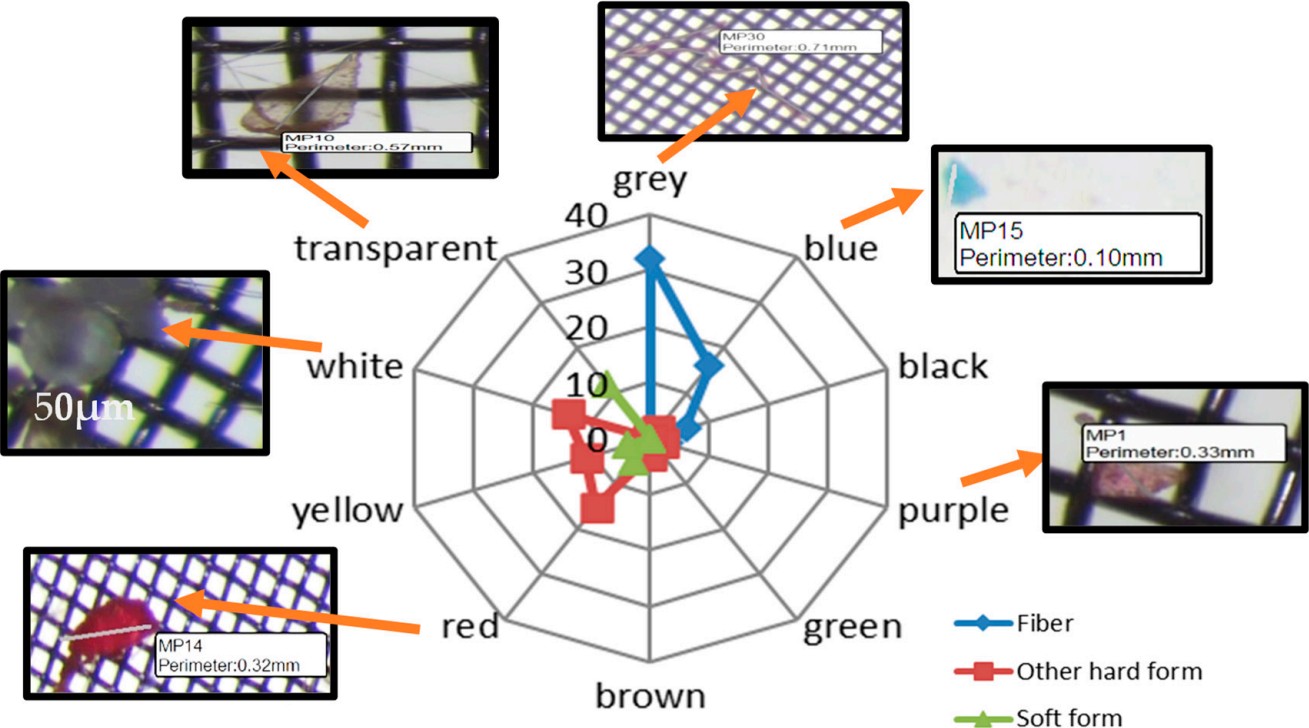

**Figure 4.** MP particles' colour distribution with several examples captured by optical microscope. The authors would like to note that captured photos exhibit slightly different colours compared with real-time observation experiences.

**Table 1.** The measured number of three identified morphotypes of MP particles depending on the filter mesh size of grey water and treated sewage samples.

| Identified Particles in Grey Water and Treated Sewage Samples | | | | | | |
|---|---|---|---|---|---|---|
| The Morphotypes MP Particles | | Shape of Fibres | | Another Hard Matrix | | Soft Matrix (Film) | |
| Applied Filter Mesh Opening Size Range, μm | | 0.7–100 | 100–300 | 0.7–100 | 100–300 | 0.7–100 | 100–300 |
| The number of MPs per 1 L in tested samples | GW$_1$ | 20 | 8 | 10 | 14 | 5 | 10 |
| | TS$_1$ | 11 | 3 | 9 | 12 | 5 | 8 |
| | GW$_2$ | 23 | 9 | 8 | 17 | 7 | 9 |
| | TS$_2$ | 11 | 5 | 6 | 16 | 5 | 6 |
| | GW$_3$ | 27 | 5 | 8 | 14 | 8 | 12 |
| | TS$_3$ | 22 | 5 | 7 | 12 | 2 | 3 |
| | GW$_4$ | 28 | 11 | 10 | 14 | 3 | 5 |
| | TS$_4$ | 19 | 10 | 8 | 13 | 2 | 1 |
| | GW$_5$ | 29 | 10 | 12 | 14 | 2 | 8 |
| | TS$_5$ | 20 | 9 | 10 | 1 | 2 | 1 |
| Total number of MP particles, pcs. | | 211 | 74 | 88 | 138 | 41 | 63 |
| Average number of MP particles, pcs. | | 21 | 7 | 9 | 14 | 4 | 6 |
| Percentage of all detected MP particles, % | | 34 | 12 | 14 | 23 | 7 | 10 |

Visual sorting revealed that detected MP particles differed in colour, shape, and number within each size group. Among all samples, on filters with a mesh size of 100 to 300 μm, a lower number of fibre-shaped particles were separated per litre: 7 (12%) compared to 14 (23%) for other solid matrices.

This confirms that the dimensions of the filter cloth affect the efficiency of the retention of microparticles. The article's authors considered this conclusion when developing a solution for improving ship wastewater treatment facilities. In the samples taken in this study, all forms of microparticles were still identified in all treated ship wastewater. Their total number was $n = 254$ particles, but one litre contains $n = 51$ particles on average. One litre of treated ship sewage contains on average $n = 23$ fibre microparticles, $n = 21$ particles of different complex matrices, and $n = 7$ particles of soft matrix or film, which can potentially end up in the marine environment.

### 3.2. Microplastics Scanning Electron Microscopy Analysis Results

The most suspicious MP particles of all morphotypes were taken from each sample for further surface characterization and elemental analysis using scanning electron microscopy (SEM) during the visual sorting procedure. Twenty-five microparticles were thoroughly tested. The results presented in Figure 5 reveal the presence of various elements on the surface of MP particles.

The captured SEM ©mages and the EDS spectrum and elemental analysis results of the detected smallest MPs (Figure 5a–f) reflect carbon©), oxygen (O), nitrogen (N), silicon (Si), titanium (Ti), zinc (Zn), sodium (Na), potassium (K) and chlorine (Cl) signals. In addition, the high proportion of fibres in samples GW3, GW4, and GW5 indicates the presence of polyamide and polyester material particles as indicated by N and O signals on the surfaces of MP particles examined by SEM/EDS.

The presence of identified Zn (Figure 5b–d) may contribute to chemical (corrosion) and tribological (erosion) processes that lead to the release of Zn from the anode material of the grey water storage tank and the surfaces of other zinc-coated materials. Titanium dioxide nanoparticles could be one of the possible forms on the surface of the tested MP samples (Figure 5b,d).In the opinion of the authors, the presence of identified Si (Figure 5b,c) could be related to reflective materials that are used in seamen's work clothes. During the sampling, authors captured photos of seamen's overall (Figure 6a,b), where damage to reflective material was observed after the first washing cycle (Figure 6a), although washing was carried out according to manufacturer instructions (Figure 6b).

SEM/EDS analysis results show that nanoparticles could be the source of the spread of inorganic nanoparticles or their modified forms in the marine environment. For example, $TiO_2$ nanoparticles (Figure 5b,d) transferred from nanomaterials can also be carriers of other environmental pollutants. Therefore, the authors would like to perform further tests with transmission electron microscopy (TEM) combined with SEM and inductively coupled plasma mass spectrometry (ICP-MS) to determine whether metals or additives are absorbed.

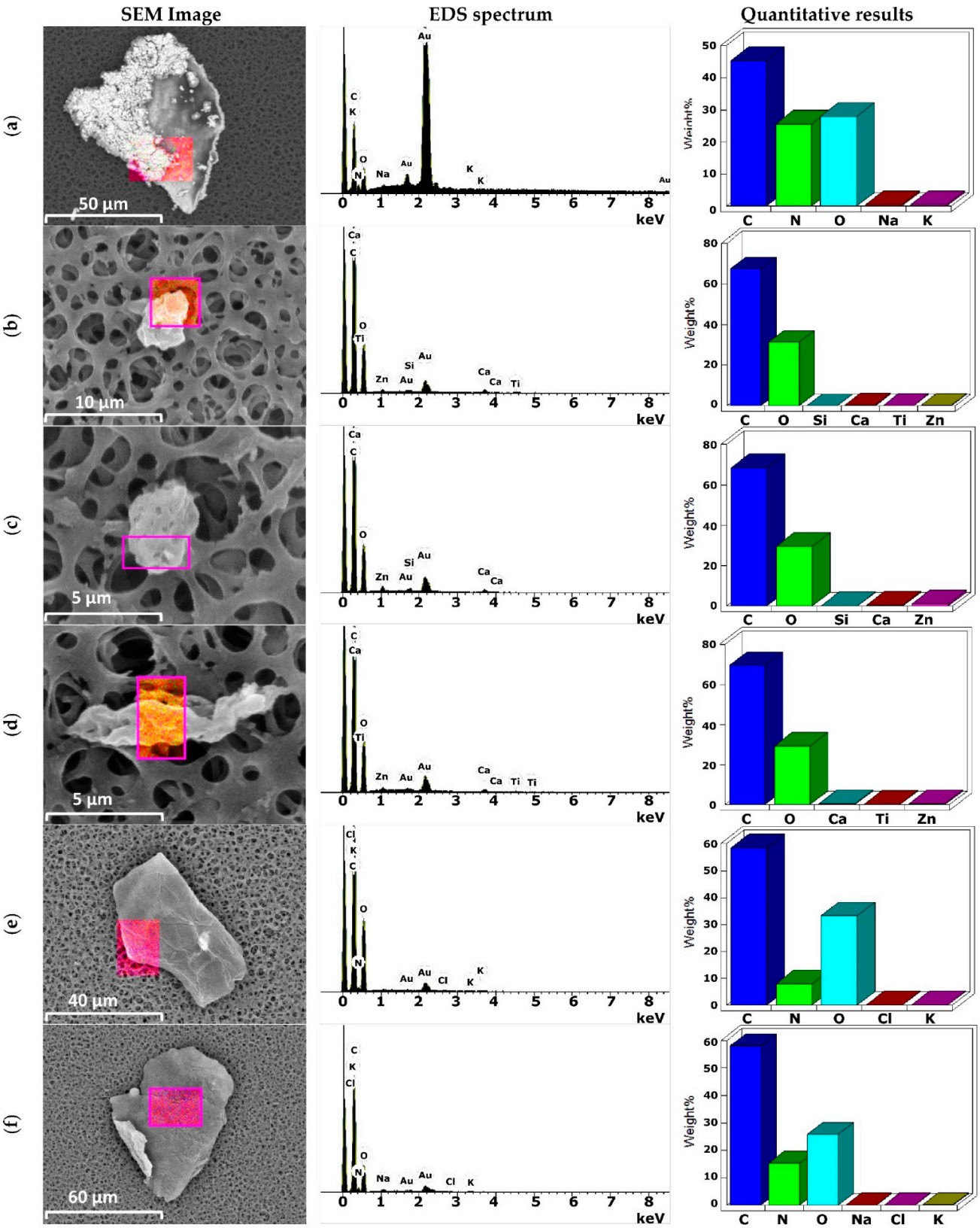

**Figure 5.** SEM images, EDS spectra and quantitative results of elemental analysis on the surfaces of possible MP particles (**a**–**f**) from grey water samples. A gold layer with a thickness of 15 nm was sputtered on the MPs' surfaces to provide higher surface conductivity for better-quality SEM images. Please see the colour version of the image to observe all marked areas in the SEM images.

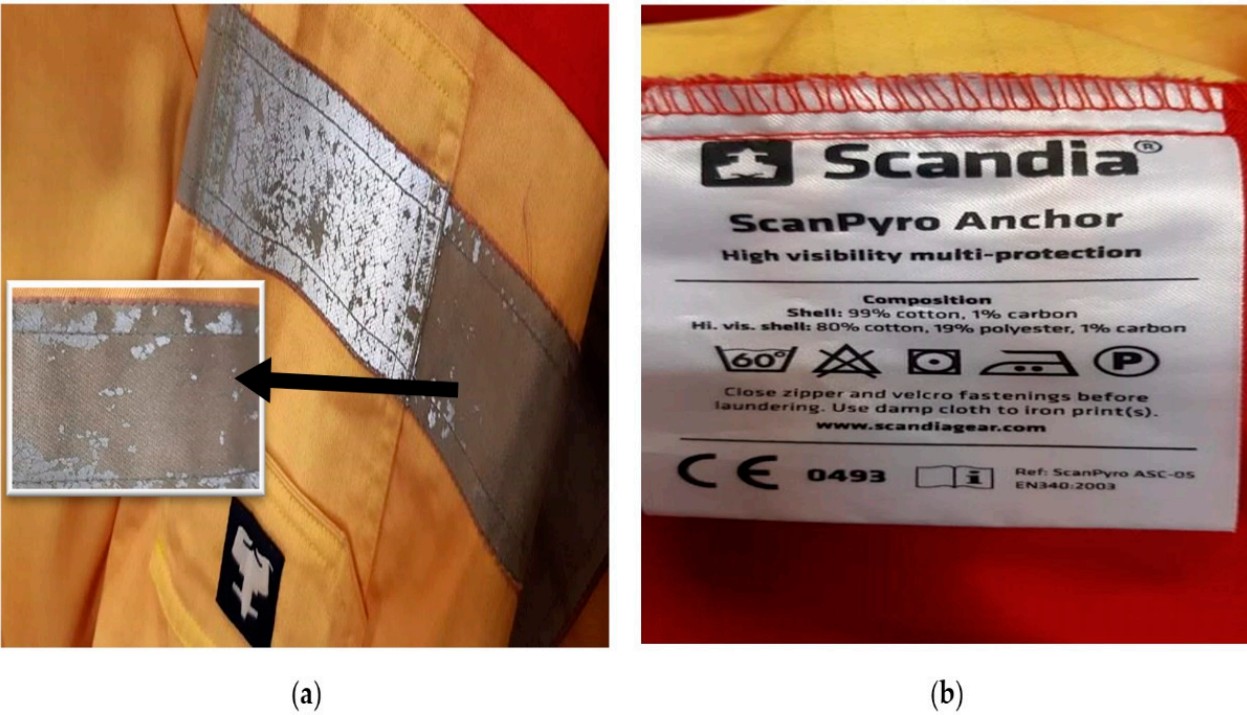

**Figure 6.** The author captured pictures onboard during sampling. The photos are of (**a**) complete damage to the reflective coating after the first washing cycle and (**b**) the ScanPyro Anchor instruction of the seaman's overalls at the top, according to which washing is carried out on board.

*3.3. Microplastics Fourier Transform Infrared (FT-IR) Spectroscopy Analysis Results*

Fifty-two microparticles covering all morphotypes from GW and ST samples were carefully examined by FT-IR spectroscopy analysis. Identification revealed plastic materials for $n = 41$ microparticles and unidentified materials for $n = 11$ microparticles. Unidentified materials referred to spectra that failed to correlate <70% or showed nonplastic material.

As nonplastic particles, FT-IR identified mainly three different colours, white ($n = 4$), yellow ($n = 2$), and brown-red ($n = 3$), irregularly shaped particles in the range from 100 μm–1.93 mm. During visual sorting, they were separated with the indication "suspected microplastics". The nonplastic white and yellow particles mainly contained stearic acid/stearate, drying oil ester acid/stearate, and castor oil. A possible source of these substances could be liquefied grease from soap or other washing liquids used by seafarers on board.

FT-IR identified the brown-red particles as alkyd resin. Alkyd resin serves as a film-forming agent in some paints. Their source could be the painted surfaces of the ship's sewage storage tanks. Other nonplastic particles identified by FT-IR were black rubber ($n = 1$) and silica ($n = 1$). The percentage of plastics identified by FT-IR was similar in both GW and ST samples. Namely, it was 65–67% of the total number of analysed microparticles. The main polymers identified in the GW and TS samples were polyester and polyamide fibres, but other hard fragments and films of low-density polyethene (LDPE) and polypropylene (PP). Among the identified MP particles, 41% were particles of various other shapes, 49% were fibres, and 10% were soft-shaped films.

Other polymers identified were polymethyl methacrylate or acrylic (PMMA), ethylene vinyl acetate (EVA), polyurethane (PU) and polystyrene (PS). Figure 7 shows the IR spectra of some typical MP particles, namely PE fragment, PP film, and polyester fibre, and the standards used for identification. Coincident peaks are highlighted for clarity.

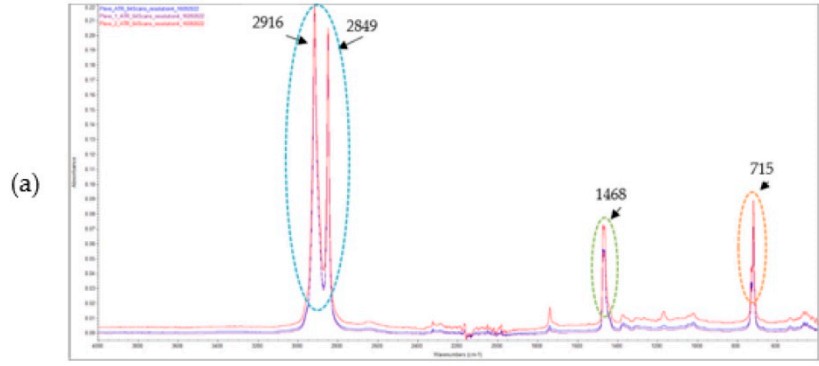

| Discernible Peaks | Peak Assignment | Confidence | Particle Definition |
|---|---|---|---|
| 2916cm⁻¹ 2849cm⁻¹ split peaks | C–H stretching, alkane | High | Polyethylene |
| 1468cm⁻¹ | CH₂ bending | | |
| 715cm⁻¹ | CH₂ rocking | | |

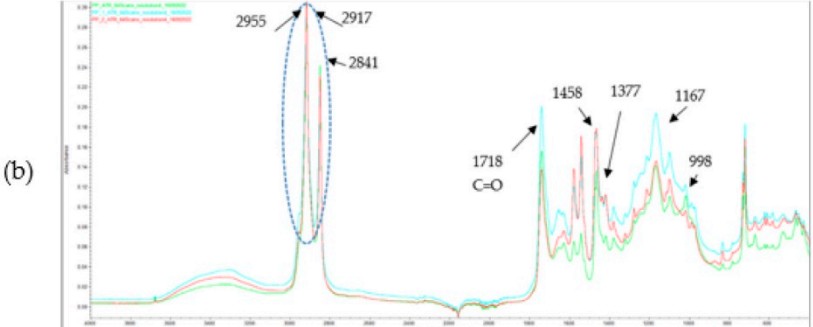

| Discernible Peaks | Peak Assignment | Confidence | Particle Definition |
|---|---|---|---|
| 2955 cm⁻¹ – 2841 cm⁻¹ split peaks | C–H stretching, alkane | High | Polypropylene |
| 1458cm⁻¹ | CH₂ bending | | |
| 1377cm⁻¹ | CH₃ bending | | |
| 1167cm⁻¹ | CH bending, CH₃ rocking | | |
| 998cm⁻¹ | C-C stretching | | |

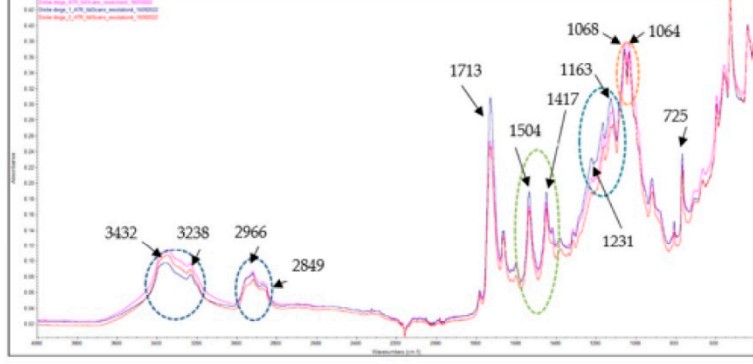

| Discernible Peaks | Peak Assignment | Confidence | Particle Definition |
|---|---|---|---|
| 3432cm⁻¹ – 3238cm⁻¹ split peaks | O-H stretching | High | Polyester |
| 2966 cm⁻¹ – 2849 cm⁻¹ split peaks | C-H stretching | | |
| 1713cm⁻¹ | C=O stretching | | |
| 1504cm⁻¹ – 1417cm⁻¹ | C–H stretching vibration of the benzene ring and the skeletal vibration of the aromatic systems | | |
| 1231cm⁻¹ – 1163cm⁻¹ split peaks | C-O stretching of the ester | | |
| 1064cm⁻¹ – 1068cm⁻¹ split peaks | C-O stretching | | |
| 725cm⁻¹ | C-H bending vibrations of the benzene rings | | |

**Figure 7.** FT-IR-ATR spectra of the most common morphological types in ship sewage samples: polyethylene (**a**), polypropylene (**b**), and a polyester fibre (**c**).

*3.4. Improvement Solution for Ship Sewage Treatment Plant*

Grey water management varies from vessel to vessel. It can be discharged into the sea untreated or mixed with black water and treated in the ship's sewage treatment plant before discharge. All treatment or delivery ashore is voluntary and per the goodwill of the shipping companies. It should be emphasized that grey water discharge needs to be regulated in most marine areas.

The results obtained in our study show that grey water and treated wastewater samples contain microplastic particles (fibres, other hard forms and soft forms or films), alkyd resins, oil and grease from food residues, detergent and soap residues, and the metals titanium and zinc. Grey water treatment reduced the number of microparticles of pollutants by approximately 29%, but 71% of them are discharged into the port waters or the sea.

The ships involved in the study were equipped with the most common sewage treatment equipment, namely biological sewage treatment equipment. This can be explained by a detailed analysis of the sewage treatment plant installed on the ships. The main components and the principle of operation are shown in the diagram in Figure 8.

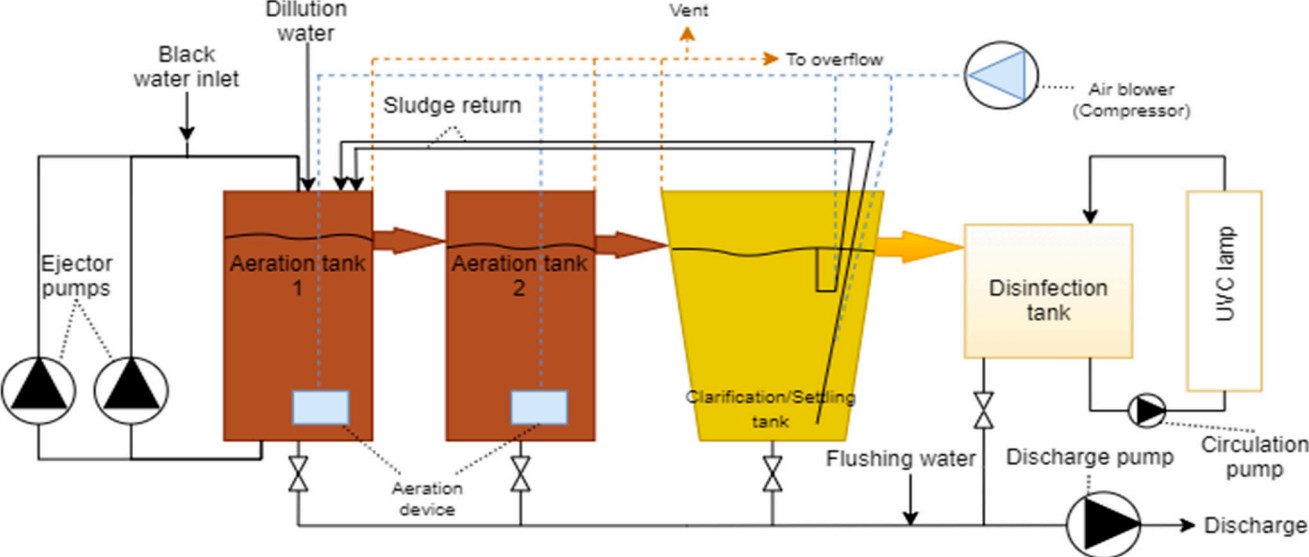

**Figure 8.** Main components and principle of operation of the biological sewage treatment plant.

Depicted (Figure 8) sewage plant consists of four tanks: two aeration tanks, a clarification or settling tank and a disinfection tank. Raw sewage enters the first aeration tank, usually through the coarse mesh screen of the mechanical treatment plant, which catches large, insoluble particles such as macro plastics parts and napkins. Then the sewage is passed to the second aeration tank. Both aeration tanks contain aerobic bacteria, which decompose organic compounds of the sewage and convert them into sludge, water, and carbon dioxide. These aeration tanks are equipped with aeration devices. Fresh air is supplied to these devices with the help of a blower to supply the aerobic bacteria with oxygen. Sewage from the second aeration tank is then passed to the clarification tank. With gravity's help, activated sludge is settled and returned to the aeration tank through the sludge return line. Clear liquid, which has settled on the top of the clarification tank, overflows into the disinfection tank. Various methods were used to disinfect the biologically treated water on the ships involved in the study, such as chloride tablets or a chlorinator, but the diagram shows a more sophisticated system with ultraviolet light installed on one of the ships.

It should be noted that the coarse mesh, whose mesh size is around 25 mm, is unable to retain MP particles in the raw sewage (see Figure 9), which further enter the aeration tanks.

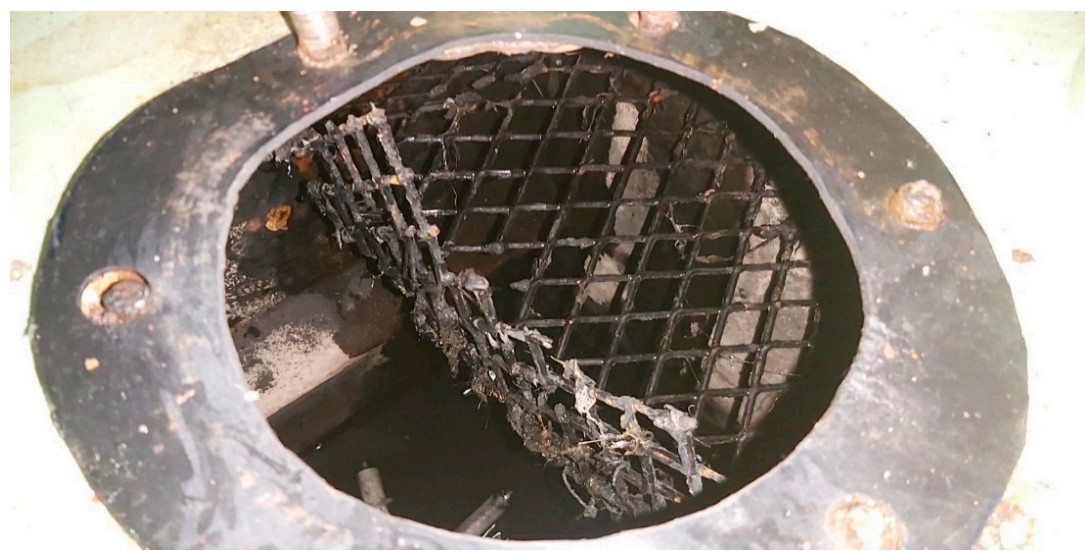

**Figure 9.** The coarse screen of mechanical purification plant.

Particles of MPs containing mixed GW and BW entering the biological treatment (aeration tanks) can become microorganism growth surfaces and support the biological treatment. As microorganisms can grow on the surface of MP particles, forming a biofilm coating, the weight of these particles will increase. Consequently, these particles are settled together with the sludge in the clarification tank by gravity and returned to the aeration tank through the sludge return line. The lighter MP particles can rise in the upper layers of water in the tank with the help of air flotation and further travel to the clarification tank. There, the water flow can quickly move to the disinfection tank, and the MP particles are discharged into the marine environment. Consequently, grey water is one of the sources of unprecedented marine pollution by microplastics. This can have a significant impact on marine protected areas and coastal cities. The small number of MP particles collected during sewage treatment in the treatment plant proves this assumption.

The solution for collecting MP particles is shown in the GW treatment diagram in Figure 10. A significant change in the GW treatment plant is provided by the filter installed in the stage of the mechanical treatment plant, which collects MP particles from the GW flow before entering the biological treatment tank.

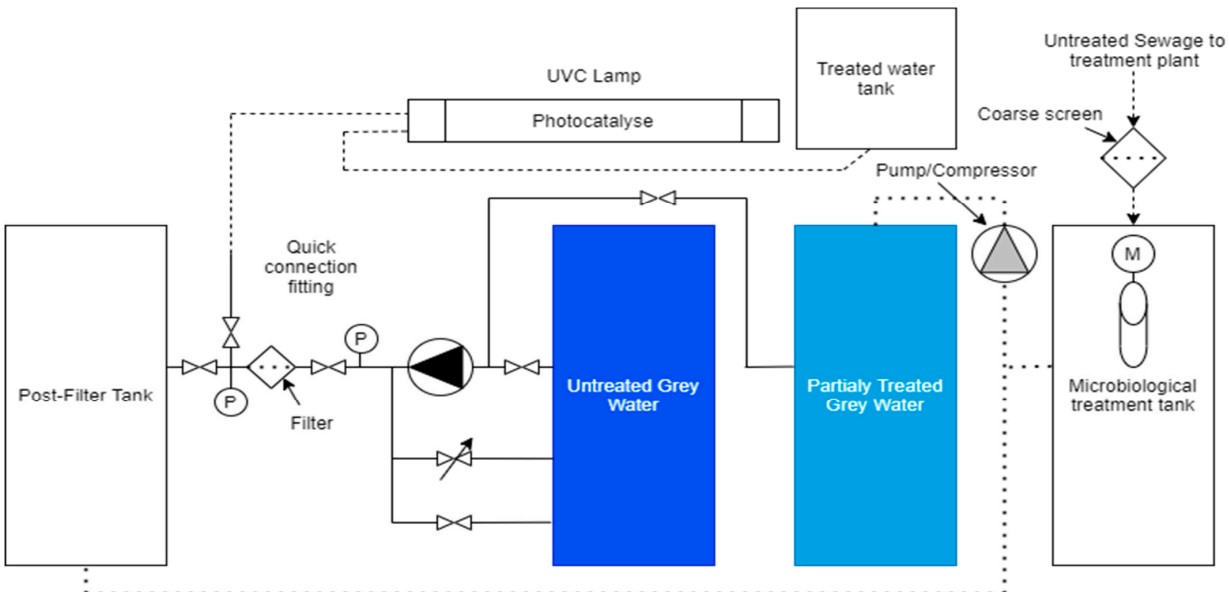

**Figure 10.** Grey water treatment plant.

The filter has an innovative design so that both filter elements and filter mesh combinations can be changed during the operation process depending on the proportion of water being filtered with microplastics or other particles of organic and inorganic origin and the degree of water turbidity, thus facilitating the operational requirements and the work of the ship's crew, ensuring the quality of the water filtering and the collection of microplastics. Glass fibre meshes are used as filtering material: $2 \times 50$ μm, $2 \times 100$ μm, $2 \times 150$ μm, perforated nonwoven five-layer glass fibre felt material, 20–40 mm peat fibres, and activated charcoal, applying the cascade method in the arrangement of filtering material in filter cartridges. The number of filter cartridges varies from eight elements to more, as does the number of filtering meshes in one cartridge element. The second filter is installed after the biological treatment tank.

There are also changes in the stage of microbiological purification. In this treatment plant, the activated sludge (ASP) method in the microbiological treatment stage is replaced by a moving bed biofilm reactor (MBBR). It is more compact than the ASP, reducing operation, maintenance, and replacement costs. After biological treatment, the purified GW is disinfected. Changes have also been made in the disinfection stage; the widely used chlorination method was replaced by semiconductor photocatalysis. A photocatalyst makes UV disinfection much more effective. This method has the potential of an alternative technology with the advantages of valuable equipment, high efficiency, energy-saving procedure for deactivating various viruses (see new coronavirus), and the breakdown of nano-sized pollutants into harmless compounds. A dual-function material consisting of $TiO_2$ nanofibers modified with low ($\leq 1\%$) silver content has excellent photocatalytic activity. The high toxicity of added Ag will give the photocatalyst the ability to kill microorganisms and pathogens. The advanced treatment plant shown in Figure 10 may be capable of treating GW to the extent that the treated water can be reused for technical purposes, thus improving the ship's eco-efficiency.

## 4. Discussion

Despite the efforts of marine and environmental scientists to improve the environmental performance of shipping from a sustainability perspective [39–41], it is still an unsolved problem [42]. Its importance is emphasized by the involvement of the IMO in the implementation of the United Nations Sustainable Development Program 2030, which defines seventeen sustainable development goals (SDGs), of which SDG 14 Conserve and sustainably use the oceans, seas, and marine resources for sustainable encompasses the protection of the marine environment [42]. For the maritime industry to ensure the implementation of SDGs, an action plan has been developed to significantly reduce all types of marine fouling from ships by 2025, including plastic waste of both macro- and microparticles [43,44]. Despite existing legislation restricting the discharge of polluted wastewater from ships, discharges still occur [31]. The results of our study indicate that ship sewage is a serious contributor to microplastic pollution (see Table 2).

**Table 2.** Total number of microparticles per L of samples and the percentage of retained particles in a sewage treatment plant.

| GW Samples | Particles/L | TS Sample | Particles/L | Percentage of All Retained Particles, % |
|:---:|:---:|:---:|:---:|:---:|
| GW$_1$ | 67 | TS$_1$ | 48 | 28 |
| GW$_2$ | 73 | TS$_2$ | 49 | 33 |
| GW$_3$ | 74 | TS$_3$ | 51 | 30 |
| GW$_4$ | 71 | TS$_4$ | 53 | 25 |
| GW$_5$ | 75 | TS$_5$ | 53 | 29 |

The total number of particles in the GW samples was 360, but after treatment in the STP, it was 254. It turns out that STP can retain about 29% of particles. The visual identification of particles confirmed that microfibers dominate as the morphological type. When the ship's crew wash their uniforms (see Figure 6) or other textiles, microfibers end up in the ship's wastewater. This is in line with evidence from studies in municipal wastewater contexts [13,44,45]. The synthetic fibres released during the washing of textiles are a significant source of microplastics, which enter aquatic ecosystems from sewage discharges through wastewater treatment plants [44,45]. In the study, the results of visual particle sorting confirm that synthetic and manufactured natural materials were present together. Using rose bengal staining, for example, out of 446 microfibers, 161 (36%) particles were recognized as nonplastic. This corresponds to the latest scientific understanding that natural fibres comprise 60–80% of the total amount of microfibers in the environment. The incorporation of natural or regenerated coloured fibres without proper chemical identification [46,47] has contributed to the increase in microplastics in both environmental matrices and organisms [48]. A study on food waste management on ships found that according to MARPOL73/78 Technical Annex V, a mixture of food waste and grey water can be discharged when sailing in the Baltic Sea more than 12 nautical miles from the nearest land [31]. This causes concern among researchers because chemical elements such as O, Si, Ti, and Zn were identified in the SEM/EDS analysis of the suspicious microparticles separated by visual sorting on the surface (see Figure 5b–d).

On the surface of the MPs particle, the chemical elements Si and O form the compound $SiO_2$. For example, chefs and food additive manufacturers also use silicon dioxide ($SiO_2$) in the particle size range of 100 to 1000 nm (or even smaller) as a food additive (E551) due to its high dispersibility. It ensures the fluidity of dry products such as milk powder, salts, and powdered sugar [49].

The detected zinc particles (Figure 5b–d) can adsorb the polymer particle material [50,51]. However, zinc oxide (ZnO, one of the possible compounds) NPs are used in packaging materials for meat products to preserve nutritional value and protect against biofouling [52]. Zinc was found in one of the highest concentrations in grey water pollutants in 44 ships [53]. The environmental monitoring of zinc in Swedish coastal waters has shown that concentrations in 19 of 32 coastal water bodies exceed the environmental quality standard (1.1 mg·L$^{-1}$) [28]. In addition, researchers have estimated that zinc emissions from ships' grey waters in the Baltic Sea could reach 2–8 tons per year [53]. However, zinc plays an essential role in human biological processes, and toxicity has been observed in various species. For example, even 5 µg·L$^{-1}$ affects the fertilization and embryo development of Baltic herring (Clupea harengus) [54]. Therefore, it is essential to reduce the release of zinc into the sea to improve the environmental condition of the Baltic Sea. The apparent presence of Ti-bearing particles (Figure 5b,d) also has several possible sources. Manufacturers typically add $TiO_2$ during the production of the polymer material as a white pigment or UV blocker [55] in cosmetics. $TiO_2$ is also a typical ingredient in hygiene products such as toothpaste [11,12,56], food packaging materials, and food products (food additive code E171) such as sweets, chewing gum, and milk powder [52].

$TiO_2$ nanoparticles are widely used in the production of packaging films because they successfully affect the contamination of food surfaces by E. coli, aerobic microorganisms. In addition, this film protects oxygen-sensitive food, helping to increase storage stability and limit losses while maintaining the quality of products such as fresh cheese, yoghurt, fresh sausages, and meat and can extend the shelf life of bread by several days [49,57]. Researchers typically identify commercially used $TiO_2$ nanoparticles as refined grains between 100 and 250 nm in diameter that can migrate from the used package due to various factors (e.g., release from the film due to mechanical stress, defects, tears, and pores) [57].

In recent years, the number of studies on the benefits of using nanoparticles in the distribution, structure, and physical/chemical properties of food and human and animal health has increased. In addition, biological and toxicokinetic studies have provided an opportunity to learn about potential hazards. Ecotoxicity data on biological effects revealed

that TiO$_2$ nanoparticles were toxic and increased mortality in animal species. This factor is associated with the inhibition of the growth of green algae (*Desmodesmus subspicatu*) [58], marine microalgae (phytoplankton; *Phaeodactylum tricornutum*) [59], and blue-green algae (*Anabaena variabilis*) as well as nitrogen fixation. Thus, releasing TiO$_2$ into the aquatic environment can potentially affect critical biogeochemical processes such as carbon and nitrogen cycling [60]. When fish consume plankton contaminated with TiO$_2$ nanoparticles, it bioaccumulates in the digestive tract of fish [61,62], where cadmium (Cd) enters together with TiO$_2$. In addition, it can adsorb on the contact surfaces of TiO$_2$ nanoparticles due to the properties of the nanoparticles, such as the high specific surface area and the active electrostatic attraction of the substance [63].

In general, FT-IR analyses confirmed that the infrared spectra of the most common polymers, polyethene, polypropylene, polyester, and acrylic fibres, correspond to the data published elsewhere [13,64,65]. The obtained infrared spectra show that $n$ = 11 microparticles (1.8%) were nonplastic microparticles, but considering this particle's anthropogenic nature and the presence of additives, it is necessary to consider the possible risk if they are released into the environment. The FT-IR spectra associated with the identified MP particles of different morphological types are shown in Figure 7. The spectrum of soft-shaped MP particles identified as polyethene (Figure 7a) have the following characteristic absorption bands: 2916 cm$^{-1}$ and 2849 cm$^{-1}$ can be attributed to C-H stretching. modes, 1468 cm$^{-1}$ to the CH$_2$ bending strain, but 715 cm$^{-1}$ to the CH$_2$ rocking strain. These findings are consistent with those reported in previous work [66,67]. For the spectrum of hard-form MP particles were identified as polypropylene (Figure 7b) due to intense absorption peaks between 2955 cm$^{-1}$ and 2841 cm$^{-1}$ (C-H stretching region), at 1458 cm$^{-1}$ (CH$_2$ bending), and 1377 cm$^{-1}$ (CH$_3$ bending) [67–69]. The band at 1715 cm$^{-1}$ (C=O stretching ketones, carboxylic acids) indicates the presence of oxygen-containing groups [70], possibly due to the abiotic oxidation of the polymer [71]. The fibre spectra (Figure 7c) show absorption at 2966 cm$^{-1}$ and 2849 cm$^{-1}$, indicating C-H stretching, but 1231 cm$^{-1}$ and 1163 cm$^{-1}$ for ester group stretching region. The band 1068 cm$^{-1}$ and 1064 cm$^{-1}$ detected CH$_2$ deformation, which is characteristic of polyester (e.g., polyethene terephthalate) [72], as it is one of the most used polymers in the textile industry [73]. The band at 725 cm$^{-1}$ is possibly due to C-H bending vibrations of the polyester benzene ring due to the surface interaction of plasma-treated polyester with the nonionic emulsion [74].

Plasma treatment is used to improve the surface properties of polymers. Glass fibre-reinforced polyester (GFRP) materials exhibit high strength-to-weight ratios and corrosion resistance and are used for various applications, particularly in civil engineering. GFRPs are often joined with similar or dissimilar materials using adhesives for such applications. However, they usually have smooth surfaces composed mainly of polyester matrix materials with low surface energies. Therefore, the adhesive joint usually requires careful surface preparation. Plasma treatment, especially at atmospheric pressure, is attractive for this application due to its environmental compatibility and high treatment efficiency without affecting the textural characteristics of the bulk material nonionic emulsion [75]. Due to the wide variety of fibres used in textile production, fibre analysis and identification are often tricky, especially when the samples are old and partially damaged [76]. However, MP particles pass through the filter sieves installed on the ship with an opening size of up to 25 mm without getting stuck. Studies reveal that plastic fragment particles can travel long distances from their source [14], absorb hydrophobic pollutants, and contain harmful additives such as phthalates and pyrene [77].

Importantly, decomposition can lead to the release of toxic compounds. Therefore, this pollution is considered a significant polluting factor for environmental safety [78]. Furthermore, microbes can quickly colonize nonbiodegradable polyethene (PE) or polybutylene terephthalate (PET) fragments and form biofilm communities. Researchers have considered these MPs from biofilm communities pathogenic, toxic, and invasive to animal species [79]. Therefore, fair practice requires the improvement of STP to retain MP particles containing GW and limit the arrival of nano- and microparticles in the marine environment.

It should be emphasized that grey water is the most significant factor in wastewater discharge on ships. The amount of grey water produced depends on the type of vessel. This can vary from 105 L (e.g., from a tanker) to 254 L (e.g., from a cruise ship) per person per day [28,31,80]. For example, in the Baltic Sea, 5.5 million tons of grey water from all types of shipping are discharged annually, with freight/vehicle transportation ships (RoPax) and cruise ships being the top contributors [53].

The information published by MEPC 74/14 (Norway) and MEPC 71/INF.22 (The Netherlands) on poor wastewater quality is a cause for concern, revealing a huge gap between the rules developed by the IMO and the reality that exists. Coliform bacteria in the STP samples were 10 million to one billion times larger than specified in the effluent standard [81]. The particular concern is cruise ships, as they carry many passengers and generate a large amount of waste compared with other ships [24,82]. It should be emphasised that ship grey water may also contain pharmaceuticals, per- and polyfluoroalkyl groups, and toxic and environmentally hazardous chemicals used to disinfect and clean during ship operations that are harmful to aquatic organisms [83,84]. Physical or chemical disinfectants may be used for disinfection. In practice, chlorine or chlorine compounds are most used onboard to kill most microorganisms. During the disinfection process, chlorine reacts with organic substances to form trihalomethane, a carcinogenic compound. Chlorination can also nitroso dimethylamines, a development associated which human cancer.

Furthermore, certain bacteria have been discovered to be immune to such chemical materials. If these substances reach the sea, they are dangerous to marine organisms and the ecosystem [85,86]. A de-chlorination step is required to keep the chlorine concentration below the 0.5 mg $L^{-1}$ limit. However, some chlorine-based sewage treatment plants do not have a de-chlorination stage [85,86].

Passenger ships have always followed security measures in the field of sanitation. However, with the COVID-19 pandemic raging, cruise passengers and crew and the cruise industry were at the forefront. Cruise vessels with many passengers and crews became COVID-19 incubators, and infections on vessels such as Diamond Princess were described as floating nightmares [82].

Water disinfection is currently a significant challenge for humanity. There is an urgent need for practical, inexpensive, environmentally friendly methods that do not promote gene exchange between bacteria. To achieve this, oxidising solid radicals are promising in this regard [87]. Studies [25,31,88,89] reveal that the wastewater treatment facilities installed on ships need to be improved because they do not perform complete and high-quality treatment of grey water.

## 5. Conclusions and Future Perspectives

This is the first study to test grey water (GW) and treated sewage (TS) samples from ships. The obtained results reveal the presence of microplastic particles (MPs) in all tested samples. After rose bengal staining, stained $n$ = 110 and partially stained $n$ = 51 particles were separated, and $n$ = 614 microparticles were studied in detail. This set of microparticles consists of $n$ = 360 microparticles separated from GW samples with a total volume of 5 L and $n$ = 254 microparticles separated from ST samples with a total volume of 5 L. The average number of separated microparticles in GW samples was $n$ = 72 particles per litre and in TS samples, $n$ = 51 particles per litre⁻. Among the 614 separated particles, the most common were fibres $n$ = 285 (46.4%), followed by other (various) hard particles $n$ = 226 (36.8%) and soft particles $n$ = 104 (16, 8%).

A total of 10 different colours formed the visually separated microparticles. Fibres were predominantly light/grey $n$ = 98 (34.4%), and could be formed from washing textiles and work overalls. Different solid-shaped particles were separated in different colours. Two colours predominated, namely red $n$ = 72 (31.9%) and white $n$ = 67 (29.8%). Soft forms or films $n$ = 15 (14.8%) was transparent. The MP particles identified ranged in size from 10 microns to 3.68 mm. The MP particles in the analysed samples may originate from seafarers'

hygiene products and intentionally or unintentionally ground food packages. In addition, MP particles were isolated from these samples, on the surface of which metals such as Ti and Zn, which are ecotoxic, were detected with SEM/EDS analysis. Regardless of whether the detected chemical elements are adsorbed or doped on the surface of the MP particles, transmission scanning analysis should be performed. FT-IR identification revealed PE, PP, polyester, polyamide, and acrylic fibres.

Our results show a worrying situation regarding the ability of ship sewage treatment plants to retain MP particles. The sewage treatment equipment of the ships involved in the study collected about 29% of microparticles, while 71% can be discharged into the port water area. On the other hand, when the ship leaves the port in the Baltic Sea further than 12 nm from the nearest land and does not violate the requirements stipulated in the MARPOL 73/78 Convention, it is allowed to discharge shredded food waste mixed with grey water. Therefore, there is a potential risk of polluting the Baltic Sea with both nutrients and MP particles, which may contain ecotoxic elements and a possible biofilm coating on PE particles with viruses (cf. COVID-19) and pathogens. This can affect the regeneration processes of the Baltic Sea. To reduce these risks, a GW treatment plant solution is offered.

Based on the study results, there is a need to change the wastewater management requirements of MARPOL 73/78 regarding grey water management. When designing and modernising technological treatment facilities, attention should be paid to the possibilities of reducing MP pollution by using environmentally friendly materials and considering the life cycle of the product and materials so that the used materials can be reused without creating additional waste for the environment. The sewage pretreatment stage is essential and requires highly effective porous filtration products. Therefore, further research will be carried out on effective microplastic collection materials and the modernisation of wastewater treatment facilities in ports.

**Author Contributions:** Conceptualization, R.K.; methodology, R.K. and I.D.; software, R.K., I.D., K.S., K.L., A.B. and R.D.; validation, R.K., I.D., K.S. and R.D.; formal analysis, R.K., investigation, R.K., I.D., K.S. and R.D.; resources, R.K.; data curation, R.K.; writing—original draft preparation, R.K.; writing—review and editing, R.K., I.D., K.S., K.L. and R.D.; visualization, R.K., I.D. and K.S.; supervision, A.B.; project administration, R.K.; funding acquisition, R.K. All authors have read and agreed to the published version of the manuscript.

**Funding:** This work has been supported by the European Regional Development Fund within the Activity 1.1.1.2 "Postdoctoral Research Aid" of the Specific Aid Objective 1.1.1 "To increase the research and innovative capacity of scientific institutions of Latvia and the ability to attract external financing, investing in human resources and infrastructure" of the Operational Programme "Growth and Employment" ("Green technology solutions for enhancing the eco-efficiency of ships for the sustainability of the Baltic Sea environment and reducing human health risks" No.1.1.1.2/VIAA/3/19/477) and by Antrex Shipping-Chartering Agency and For-warding ser.

**Institutional Review Board Statement:** Not applicable.

**Informed Consent Statement:** Not applicable.

**Data Availability Statement:** Not applicable.

**Acknowledgments:** This work has been supported by the European Regional Development Fund within the postdoctoral (postdoc) project "Green technology solutions for enhancing the eco-efficiency of ships for the sustainability of the Baltic Sea environment and reducing human health risks" No.1.1.1.2/VIAA/3/19/477. The authors would like to thank the team of Riga Technical University (Latvia) and shipping companies for their support in the implementation of the study. The authors would also like to mention the support from D. Loca (Riga Technical University, Latvia).

**Conflicts of Interest:** The authors declare no conflict of interest. The funders had no role in the design of the study; in the collection, analyses, or interpretation of data; in the writing of the manuscript; or in the decision to publish the results.

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
