# Peer review of "Microplastics in Ship Sewage and Solutions to Limit Their Spread: A Case Study"

_water, doi:10.3390/w14223701_

Round 1
Reviewer 1 Report
The manuscript is very interesting because it deals about a new kind, of marine pollution by microplastics caused by shipping traffic. By this way, the issues treated in the article could be useful for a sound management of marine environment. Finally, the scientific topics coming out from the paper are of great interest for international readers.

Author Response
Thank you for reviewing our article manuscript Microplastics in ship sewage and solutions to limit its spread: A case study, we are glad that You consider our topic interesting and international.
Please see the attachment.

Reviewer 2 Report
The authors conducted an interesting and relevant scientific study. The manuscript is well written and formatted.
Minor comments:
1. Page 16, lines 434-439: the discussion of the applicability of photocatalysis should be reconsidered as this process is basically an advanced oxidation process rather than a disinfection process.
2. Page 17, line 486: „the detected zinc particles (Figure 6 b, c and d) containing particles can adsorb…“ most likely here should be (Figure 5 b, c and d). Please correct.
Author Response
Thank you for reviewing our article manuscript Microplastics in ship sewage and solutions to limit its spread: A case study. We are glad that You have considered our article manuscript well-written and formatted. Also, it flatters that You have considered that the authors have conducted an interesting and relevant scientific study.
Answering to comment section:
|
1. Page 16, lines 434-439: the discussion of the applicability of photocatalysis should be reconsidered as this process is basically an advanced oxidation process rather than a disinfection process.
|
However, the article’s authors have directly indicated that photocatalysis improves UV radiation disinfection because the oxidation-reduction process is ensured when UV radiation acts on the photocatalyst material. It generates hydroxyl radicals (OH*) and superoxide anions (O2-), which have a destructive effect on biological and chemical water pollutants and disinfect it. 1) Photocatalyst makes UV disinfection much more effective. This method is the potential of an alternative technology with the advantages of valuable equipment, high efficiency, and energy-saving procedure for deactivating various viruses (see new coronavirus) and the breakdown of nano-sized pollutants into harmless compounds. A dual-functional material consisting of TiO2 nanofibers modified with low (≤1%) silver content has excellent photocatalytic activity. The high toxicity of added Ag will give the photocatalyst the ability to kill microorganisms and pathogens. The advanced treatment plant shown in Figure 10 may be capable of treating GW to the extent that the treated water can be reused for technical purposes, thus improving the ship's eco-efficiency. |
|
2. Page 17, line 486: „the detected zinc particles (Figure 6 b, can d) containing particles can adsorb…“most likely here should be (Figure 5 b, c and d). Please correct. |
Your indication of “the detected zinc particles (Figure 6 b, c, and d) containing particles “has been corrected. |
Reviewer 3 Report
This study monitored information on the content of MPs in GW and treated sewage (TS) samples and provided a possible solution for the collection of MPs in the ship's sewage treatment plant. The study results support the hypothesis about GW and TS as important sources of MP pollution. I have no doubt about sampling, processing and analysis methods. The topic is suitable for the journal and the information is very useful. There are some minor revisions necessary to make this manuscript suitable for publish in the journal.
Line 314-323 Did authors tested the EDS spectra of the polymeric materials from seaman’s uniform?
Line 364 The authors proposed a solution of ship treatment plant for removal of microplastic. However, there are no conclusive results to prove the reliability of this solution. The author needs to explain this issue.
Figure 1, 6, and 9 I don't think these figures are suitable for the main text.
Figure 5 and 7 The image clarity needs to be improved.
References The journal names should be unified with others.
Author Response
Thank you for reviewing our article manuscript Microplastics in ship sewage and solutions to limit its spread: A case study. We are glad to hear, that You have considered that our article manuscript topic is suitable for the journal and the information is very useful.
Answering to comment section:
|
Line 314-323 Did the authors test the EDS spectra of the polymeric materials from the seaman’s uniform?
|
The authors of the paper did not conduct EDS of the material of the sailor uniform because in the paper, EDS of the particles detected by GW were studied. Researchers in future studies will certainly do this. |
|
Line 364 The authors proposed a solution of the ship treatment plan for the removal of microplastic. However, there are no conclusive results to prove the reliability of this solution. The author needs to explain this issue.
|
The authors present this paper's conceptual model of GW purification. Experimental studies have been carried out for each device offered in the model. Right now, the entire proposed system is being tested—the authors of the article plan to publish the obtained results in the following article. |
|
References The journal names should be unified with others. |
Reference design style has been improved |